# Exogenous Phytosulfokine α (PSKα) Alleviates Chilling Injury of Kiwifruit by Regulating Ca^2+^ and Protein Kinase-Mediated Reactive Oxygen Species Metabolism

**DOI:** 10.3390/foods12234196

**Published:** 2023-11-21

**Authors:** Di Wang, Xueyan Ren, Lingkui Meng, Renyu Zheng, Dong Li, Qingjun Kong

**Affiliations:** 1Xi’an Key Laboratory of Characteristic Fruit Storage and Preservation, Shaanxi Engineering Laboratory of Food Green Processing and Safety Control, College of Food Engineering and Nutritional Science, Shaanxi Normal University, Xi’an 710119, China; wangdi237@163.com (D.W.); 15030882625@163.com (L.M.);; 2College of Biosystems Engineering and Food Science, Zhejiang University, Hangzhou 310058, China

**Keywords:** chilling injury, PSKα, ROS metabolism, Ca^2+^ signaling, protein kinase, kiwifruit

## Abstract

Kiwifruit fruit stored at low temperatures are susceptible to chilling injury, leading to rapid softening, which therefore affects storage and marketing. The effect of 150 nM mL^−1^ of exogenous phytosulfokine α (PSKα) on reactive oxygen species (ROS) metabolism, Ca^2+^ signaling, and signal-transducing MAPK in kiwifruit, stored at 0 °C for 60 days, was investigated. The results demonstrated that PSKα treatment effectively alleviated chilling injury in kiwifruit, with a 15% reduction in damage compared to the control on day 60. In addition, PSKα enhanced the activities and gene expression levels of superoxide dismutase (SOD), catalase (CAT), ascorbate peroxidase (APX), glutathione reductase (GR), Ca^2+^−ATPase, and mitogen−activated protein kinase (MAPK). In contrast, the activities and gene expression levels of NADPH oxidase (NOX) were inhibited, leading to a lower accumulation of O_2_^−^ and H_2_O_2_, which were 47.2% and 42.2% lower than those in the control at the end of storage, respectively. Furthermore, PSKα treatment enhanced the calmodulin (CaM) content of kiwifruit, which was 1.41 times that of the control on day 50. These results indicate that PSKα can mitigate chilling injury and softening of kiwifruit by inhibiting the accumulation of ROS, increasing antioxidant capacity by inducing antioxidant enzymes, activating Ca^2+^ signaling, and responding to MAPK protein kinase. The present results provide evidence that exogenous PSKα may be taken for a hopeful treatment in alleviating chilling injury and maintaining the quality of kiwifruit.

## 1. Introduction

Kiwifruit (*Actinidia deliciosa* cv.) is an economically important fruit globally for its good taste and its richness in nutrients such as vitamin C. Currently, China has become the world’s largest producer and consumer of kiwifruit [1]. However, kiwifruit is prone to rapid softening and rotting after harvesting, resulting in a short shelf-life and a significant reduction in its commercial value [2]. While low-temperature storage can effectively inhibit kiwifruit softening and prolong its life, it can also induce chilling injury (CI). Chilling injury can make the fruit vulnerable to rotting and deterioration [2] and can significantly reduce the economic value of the kiwifruit. Hence, developing a promising potential practical method for alleviating chilling injury in kiwifruit has become a key focus for numerous researchers.

Phytosulfokine−α (PSKα) is a peptide endogenous plant hormone that plays an important role in plant growth and development, symbiotic interactions, and defense responses [3]. Recent research has shown that PSKα can enhance chilling tolerance and reduce the occurrence of chilling injury in horticultural products [3]. In strawberry fruit, PSKα delayed senescence and reduced the decay of strawberry fruit during low temperature storage by regulating energy metabolism [4]. It has been reported that exogenous PSKα increased intracellular adenosine triphosphate (ATP) levels to enhance chilling tolerance in loquat fruit [5]. In addition, our previous study proved that PSKα could alleviate chilling injury in bananas by inducing the accumulation of NO, proline, and gamma-aminobutyric acid (GABA) [3].

Reactive oxygen species (ROS) burst is one of the primary responses of plants against abiotic stress. However, excessive reactive oxygen species causes cellular oxidative damage and accelerates cellular senescence; therefore, the ROS scavenging system in plants is activated in response to reducing oxidative stress [6]. The ROS scavenging system includes antioxidant substances (ascorbic acid (AsA), glutathione (GSH), polyphenols, flavonoids) and antioxidant enzymes (superoxide dismutase (SOD), catalase (CAT), ascorbate peroxidase (APX), glutathione reductase (GR)) [7]. O_2_^−^ is converted to H_2_O_2_ and O_2_ under the catalysis of SOD. CAT further converts H_2_O_2_ to H_2_O and O_2_. APX and GR maintain intracellular ROS homeostasis in plants by participating in the AsA–GSH pathway [6]. The ROS−scavenging system has been proven to have a positive effect on chilling injury in many horticultural crops, such as bell peppers [8], okra pods [9], mango fruit [10], and blackberries [11]. Meanwhile, the antioxidative effects of PSKα have been reported in broccoli florets [12] and loquats [5]. However, few studies have been performed on the effect of PSKα on ROS metabolism and interaction with other signal molecules in horticultural products.

Ca^2+^ is involved in the regulation of chilling response pathways as an important secondary signaling molecule in plant signal transduction pathways. Intracellular Ca^2+^ transmits chilling signals by interacting with calmodulin (CaM), calcineurin B-like proteins (CBLs), CBL−interacting protein kinases (CIPKs), and calcium-dependent protein kinase (CDPK). CaM is a highly conserved Ca^2+^ receptor protein that is ubiquitous in eukaryotic organisms [13]. Upon the binding of CaM to the second messenger Ca^2+^, Ca^2+^ can influence cellular function by regulating the activity of various enzymes in the cell. Activated CaM can bind to different substances to produce its effects. For example, it binds directly to target enzymes and regulates their activities by inducing active conformation of the target enzymes, such as Ca^2+^−ATPase, NAD kinase, etc. [13]. Studies have shown that Ca^2+^ was closely related to plant chilling stress resistance [14]. It has also been reported that alleviated chilling injury was observed when Ca^2+^ effluxes were reduced in bananas [15]. In *Dalbergia odorifera*, an increase in Ca^2+^ content induced the accumulation of proline and sugar to defend against cold stress [16]. In addition, Ca^2+^−CaM might be involved in signal transduction events leading to proline accumulation in *Jatropha curcas* seedlings under chilling stress [17].

Mitogen-activated protein kinase cascade (MAPK) is a vital pathway in intracellular signal transduction. MAPK signaling cascades are essential for various physiological responses, for instance, plant growth and development, stress response, etc. [18]. The MAPK signaling cascade involves three types of proteins: MAPK kinase kinase (MAPKKK), MAPK kinase (MAPKK), and MAPK [19]. It has been confirmed that MAPK plays a significant role in cold response in plants [20]. At low temperatures, the expression of MPK3 was rapidly induced in *Arabidopsis*, leading to the activation of the kinase activities of MPK4 and MPK6 [20]. In tobacco, the over-expression of ZmMPK17 contributed to proline accumulation, which positively regulated resistance against chilling stress [21]. Additionally, it has been found that the Ca^2+^/CaM receptor protein kinase CRLK1 plays a key role in the MEKK1−MKK2−MPK4/MPK6 cascade signaling system activated by low temperatures [18].

At present, the majority of research studies of PSKα on chilling injury have concentrated on the effect of metabolisms on energy, proline, and NO. However, there have been few studies focused on the effect of PSKα on chilling injury and ROS metabolisms, Ca^2+^ signaling, and MAPK protein kinase in kiwifruit during cold storage. To get a further understanding of the role of exogenous PSKα, 150 nM PSKα was applied, and the purpose of this study was to investigate the impact of PSKα on ROS metabolism, Ca^2+^ signaling, and MAPK protein kinase, to illustrate the fundamental action mechanisms of PSKα in alleviating the chilling injury of kiwifruit.

## 2. Materials and Methods

### 2.1. Chemicals and Reagents

PSKα (≥95%) was purchased from Pepmic Co., Ltd. (Suzhou, China). All other chemicals were of analytical reagents grade and were purchased from Sinopharm Chemical Reagent Co., Ltd. (Shanghai, China).

### 2.2. Fruit Material and Treatments

Kiwifruit (*Actinidia delicious* cv. Cuixiang) at optimum maturity (total soluble solid content of 6.5 °Brix ± 7.5 °Brix, determined according to normal industry criteria) were harvested from a kiwifruit orchard in Shaanxi province and transported to the Postharvest Laboratory of Shaanxi Normal University on the same day. Kiwifruit of uniform size, and without disease, insect pests, and mechanical injury, were selected for this study.

Pre-experiments were set up with four PSKα treatment concentrations, 50, 75, 150, and 300 nmol L^−1^, and a distilled water treatment was used as a control; 150 nmol L^−1^ was determined as the optimal treatment concentration. Then, the fruits were soaked with 150 nmol L^−1^ of PSKα solution for 10 min. The initial values were determined on the day of storage; thereafter, 150 fruits were taken from the treatment and control groups at 10-day intervals during the storage period for the determination of quality indexes, with three replicates of 50 fruits per group.

After the fruit surface was dried naturally, the fruit were placed in cold storage at 0 ± 0.5 °C, and relative humidity (RH) was 90–95%. During the storage period, 15 fruits were taken every 10 d for the determination of firmness, TSS, TA, MDA content, membrane permeability, and reactive oxygen species content. At the same time, samples from the equatorial region of the fruit were frozen with liquid nitrogen and stored in a refrigerator at −80 °C for the determination of other indexes. Each index was repeated 3 times, and the average value was taken.

### 2.3. Assay of Firmness and Color Changes (L* Values)

The firmness of the kiwifruit was determined as described by Tylewicz et al. [22] using a texture analyzer (Instron LTD. High Wycombe, Bucks, UK) with a 5 mm probe, when the penetration speed was 1 mm s^−1^, and the test distance was 5 mm. This test evaluates fruit firmness. The maximum force needed to puncture the skin (expressed in N) was used to represent fruit firmness.

For indicating browning, the L-values were considered. If the authors have recorded L, a, and b values, they can calculate the browning index using the conversion formula as given in a reference: Browning Index (BI) = [100 (x − 0.31)]/0.17, where x = (a* + 1.75L*)/(5.645L* + a* − 0.3012b*), according to Ruangchakpet and Sajjaanantakul [23].

### 2.4. Total Soluble Solids (TSS) and Titratable Acidity (TA)

To assay TSS, the kiwifruit were juiced, and the TSS was measured using a refractometer (PR−101α, ATAGO Co., Ltd., Tokyo, Japan) and expressed as °Brix.

The TA content of the kiwifruit was determined by the method of Wang et al. [3]. Approximately 10 g of fruit tissue was ground in 5 mL of distilled water and centrifuged at 10,000× *g* for 10 min. Distilled water was added to the supernatant to 10 mL, and the mixture was boiled for 5 min to remove carbon dioxide (CO_2_) and then titrated with 0.1 M NaOH solution. TA content was expressed as a %.

### 2.5. Chilling Injury (CI) Incidence and Chilling Injury Index (CII)

The incidence of chilling injury and chilling injury index (CII) were measured with reference to the method of Wang et al. [3] and slightly modified.
Chilling injury incidence (%) = (number of chilling injured fruit/number of total fruit) × 100% 

Chilling injury index (CII) was as follows:CII = ∑ (chilling injury scales × number of chilling injured fruit)/(the highest scale × number of total fruit)

Chilling injury degree is divided into 5 scales, “0”: smooth skin without chilling injury symptom; “1”: <20% chilling injury area; “2”: 20~40% chilling injury area; “3”: 40~60% chilling injury area; “4”: >60% chilling injury area.

### 2.6. Measurement of Electrical Leakage and Malondialdehyde (MDA) Content

According to the method of Wang et al. [3], the MDA content was detected. A 2 g sample of fruit tissue was mixed with 5 mL trichloroacetic acid (TCA) solution (100 g L^−1^), homogenized, and centrifuged at 10,000× *g* for 15 min. Then, 3 mL of supernatant was collected and mixed with 7.2 g L^−1^ of TCA solution; the mixture was sealed with plastic wrap and boiled for 20 min. The absorbance was measured at 600 nm, 532 nm, and 450 nm after cooling.

The electrolyte leakage was determined according to the method of Wang et al. [3]. Slices with a (1 cm^2^) diameter were cut from the equatorial region and soaked in deionized water at 25 °C for 30 min. The initial electrolyte leakage (L0) was determined using the conductivity meter, and the electrolyte leakage (L1) was determined after boiling for 30 min and cooling to room temperature. The total electrolyte leakage is expressed as a ratio of L0 to L1.

### 2.7. The Rate of O_2_^−^ Production and H_2_O_2_ Content Assay

The H_2_O_2_ content was determined by a H_2_O_2_ assay kit (Nanjing Jiancheng Bioengineering Research Institute, Nanjing, China) with the instruction of the test kit. H_2_O_2_ content was expressed based on fresh weight as 10^−6^ mol g^−1^.

The production of O_2_^−^ was determined by a superoxide anion assay kit (Nanjing Jiancheng Bioengineering Research Institute, Nanjing, China) with the instruction of the test kit. The production rate of O_2_^−^ was expressed based on fresh weight as 10^−9^ mol g^−1^ min^−1^.

### 2.8. Detection of Calmodulin (CaM) Content

The CaM content was determined by a CaM ELISA kit (Shanghai Sailing Biological Institute, Shanghai, China) with the instruction of the test kit. CaM content was expressed based on fresh weight as 10^−9^ g^−1^.

### 2.9. Ca^2+^−ATPase and MAPK Activities

The Ca^2+^−ATPase activity was measured by a Ca^2+^−ATPase test kit (Nanjing Jiancheng Bioengineering Research Institute, Nanjing, China) with the instruction of the test kit. Ca^2+^−ATPase activity was expressed based on protein as μmol Pi 10^−3^ mg^−1^ h^−1^.

The MAPK activity was determined by a MAPK ELISA kit (Shanghai Sailing Biological Institute, Shanghai, China) with the instruction of the test kit. MAPK activity was expressed based on fresh weight as △OD_450_ g^−1^.

### 2.10. Detection of NOX, SOD, CAT, APX, and GR Activities

The NOX activity was determined by a NOX test kit (Nanjing Jiancheng Bioengineering Research Institute, Nanjing, China) with the instruction of the test kit. NOX activity was expressed based on fresh weight as △OD_470_ g^−1^.

The SOD activity was determined by a SOD test kit (Nanjing Jiancheng Bioengineering Research Institute, Nanjing, China) with the instruction of the test kit. SOD activity was expressed based on fresh weight as △OD_560_ g^−1^.

The CAT activity was determined by a CAT test kit (Nanjing Jiancheng Bioengineering Research Institute, Nanjing, China) with the instruction of the test kit. CAT activity was expressed based on fresh weight as △OD_240_ g^−1^.

The APX activity was determined by an APX test kit (Nanjing Jiancheng Bioengineering Research Institute, Nanjing, China) with the instruction of the test kit. APX activity was expressed based on fresh weight as △OD_290_ g^−1^.

The GR activity was determined by a GR test kit (Nanjing Jiancheng Bioengineering Research Institute, Nanjing, China) with the instruction of the test kit. GR activity was expressed based on fresh weight as △OD_340_ g^−1^.

The Coomassie brilliant blue method was used to quantify the protein content [24].

### 2.11. Statistical Analysis

Three biological replicate samples were included for each treatment. Data were analyzed using the SPSS11.0 software package (SPSS Inc., Chicago, IL, USA). A one−way analysis of variance (ANOVA) was performed, followed by Duncan’s post hoc test, where significance was set at a *p*-value < 0.05. Result data are presented as mean ± standard deviation.

## 3. Results

### 3.1. Firmness, Color Change, TSS, and TA Content

The firmness changes of kiwifruit with PSKα−treated and control groups were presented in Figure 1A. During storage, a decrease in firmness was observed in both groups. However, compared to the control group, PSKα treatment delayed the decrease in firmness. At the end of storage, the firmness of the PSKα−treated group was about 1.95 times greater than that of the control group.

Figure 1B shows the color changes (L* values) of kiwifruit during storage. The L* values of kiwifruit fruit flesh changed more significantly during storage, and a continuous decrease was observed in both the PSKα−treated and control groups with the storage time extended. At day 60, the control L* values were significantly lower than those of the PSKα−treated group (*p* < 0.05).

Figure 1C,D show the changes in the TSS and TA of kiwifruit in the PSKα−treated and control groups, respectively. The results showed that the TSS content of fruit in both groups increased during storage (Figure 1C). Compared to the control group, PSKα treatment delayed the increase of the TSS content. In the earlier storage period, the TSS content of the PSKα−treated group was significantly higher than that of the control group (*p* < 0.05). On the 30th day, the TSS content of the PSKα−treated group increased by 30.8% compared to that of the control group (Figure 1C).

The TA content of the PSKα−treated and control groups showed a decreasing trend throughout the storage period (Figure 1D). Compared to the control group, the PSKα treatment delayed the decline of TA. At the end of storage, the TA content of the PSKα−treated group was 18.2% higher than that of the control (Figure 1D).

### 3.2. Chilling Injury Incidence, Chilling Injury Index, MDA, and Electrolyte Leakage

Figure 2A indicates that chilling injury symptoms began to appear in both the control and PSKα−treated groups on day 50 and increased with storage time. Notably, PSKα treatment significantly reduced the chilling injury symptoms of the kiwifruit throughout the entire storage period. At the end of the storage period, the incidence of chilling injury in the PSKα−treated group was 15% lower than that of the control group.

The chilling injury index of both the PSKα−treated and control groups increased from day 20 (Figure 2B). At 60 days, the chilling injury index of the PSKα−treated fruit was 2.17, which was 24.9% lower than that of the control fruit.

The MDA content and the cell membrane permeability of the kiwifruit increased with storage time, and the MDA content and electrolyte leakage of the control group were significantly higher than those of the PSKα−treated group throughout the storage period (Figure 2C,D).

### 3.3. H_2_O_2_ Content and Generation Rate of O_2_^−^·

As is shown in Figure 3A, the content of H_2_O_2_ increased in both groups throughout the whole storage period. The H_2_O_2_ content in the control group peaked at 50 d, which was 42.2% higher than the PSKα−treated group. In contrast, the maximum H_2_O_2_ content in the PSKα-treated group occurred at 60 d, a delay compared to the control group. The production of O_2_^−^ increased significantly in the control group compared to the PSKα-treated group during the middle and later periods of storage. Both groups displayed a peak in O_2_^−^ production at 60 d, with the PSKα−treated group producing 1.47 times the amount of O_2_^−^ compared to the control group (Figure 3B).

### 3.4. Activities of NOX, SOD, CAT, APX, and GR

Compared to PSKα−treated group, NOX activities increased in the control group (Figure 4A). Figure 4A also shows there was no significant difference in the change of NOX activity between the control and PSKα−treated groups at the earlier phase of storage. However, after this period, NOX activity continued to increase in the control group. By the 40th day, the NOX activity of the PSKα−treated group was significantly lower than that of the control group.

As shown in Figure 4B, the SOD activity increased in both the PSKα−treated and control groups, peaking at 30 days. At this point, the SOD activity in the PSKα−treated group was 1.2 times higher than that of the control group. The CAT activity of the PSKα−treated group increased initially and then decreased (Figure 4C). At 40 days, the CAT activity of the PSKα−treated group peaked, being 1.42 times higher than that of the control group. As shown in Figure 4D, the PSKα−treated treatment induced a rapid increase of APX activity, reaching its peak at 20 days. At the end of the storage period, the APX activity of the PSKα−treated group was 17.1% higher than that of the control group (Figure 4D). As is indicated in Figure 4E, the GR activity of the PSKα−treated group was lower than that of the control group during the first 20 days but continued to increase thereafter. The PSKα−treated treatment-induced GR activity increased in the later stage, which was significantly higher than of the control on days 40 and 50 (*p* < 0.05).

### 3.5. CaM Content and Activities of Ca^2+^−ATPase and MAPK

Treatment with PSKα induced a significant accumulation of CaM in kiwifruit during the whole storage period, and CaM content peaked on day 40, exhibiting a 1.41-fold increase compared to the control group (Figure 5A).

During the early stage, Ca^2+^−ATPase activity increased gradually, with no significant difference observed between the control and PSKα−treated groups. On the 40th day, Ca^2+^−ATPase activity was significantly higher in the treated group compared to the control group (*p* < 0.05). At the end of the storage period (day 60), Ca^2+^−ATPase activity in the PSKα group was 1.4 times higher than that of the control group (Figure 5B).

Compared to the control group, the MAPK activities of the PSKα group increased significantly during storage (*p* < 0.05). The PSKα−treated group exhibited two peaks in MAPK activity on days 20 and 50, with levels being 9.3% and 15.7% higher than those of the control group (Figure 5C), respectively.

## 4. Discussion

Kiwifruit is susceptible to chilling injury during low-temperature storage, which severely affects its quality and commercial value [2]. A number of studies have shown that PSKα effectively maintained the quality of fruits and vegetables while extending their shelf life. PSKα has been used in alleviating chilling injury in various horticultural products [4,5,7,19]. Previously, we demonstrated the positive effect of PSKα in mitigating chilling injury in bananas [3]. Hence, some researchers have speculated that PSKα has great potential for controlling chilling injury and maintaining the quality of fruits and vegetables.

One of the main manifestations of chilling injury in kiwifruit is pulp browning [2]. The changes in fruit pulp color were characterized by L* values, with lower L* values representing lower brightness and more severe flesh browning. This study showed that PSKα treatment inhibited browning, maintained higher soluble solids, and delayed the decrease of TA in kiwifruit during storage at 0 °C (Figure 1B–D). Meanwhile, compared to the control group, PSKα maintained fruit firmness and improved its storability (Figure 1A). In the present study, CI symptoms were observed exhibited at day 20 and characterized by water-soaked pulp, tissue lignification, uneven softening, and browning of the exocarp and pulp. In addition, Figure 2 shows that during storage, the MDA content, electrical leakage, and cell membrane permeability continuously increased. PSKα treatment reduced the CI index and CI incidence in kiwifruit; meanwhile, membrane integrity was maintained in the treatment group (Figure 2A,B). This finding is consistent with previously published studies on the banana [3], strawberry [25], broccoli [26], and loquat [5], which have shown that PSKα treatment can alleviate CI symptoms and maintain crop quality.

Massive production of ROS was induced by low temperature in plants, including O_2_^−^ and H_2_O_2_, which was considered one of the earliest responses to chilling stress [26]. Many studies showed that the accumulation of ROS is positively associated with chilling injury in crops. Generally, high levels of H_2_O_2_ and O_2_^−^ could lead to severe CI, which has been found in sweet pepper, [27] tomato fruit [28], okra [29], cucumber [30], and pokan fruit [31]. In the present study, the production rate of O_2_^−^ and H_2_O_2_ content increased in both PSKα-treated and untreated fruit as the storage period was extended (Figure 2). The kiwifruit treated with PSKα could inhibit accumulation of O_2_·^−^ and H_2_O_2_ in the early stage, which might be essential for chilling injury prevention and quality maintenance (Figure 2). A previous study has shown that PSKα delayed the senescence of broccoli florets by avoiding ROS accumulation, which is consist with present results [26].

NADPH oxidase (NOX) was thought to be the key enzyme in the generation of ROS in plants under normal and stress conditions. On the plasma membrane, NOX catalyzes O_2_ into O_2_^−^· with NADPH as an electron donor [7]. The present study found a significant decrease in NOX activity in PSKα-treated kiwifruit (Figure 3), aligning with the changes in ROS levels. In tomato fruit, NOX activity was found to increase during low temperature in the early stage, which can be responsible for H_2_O_2_ accumulation [31]. Similar results were reported in mango fruit [32]. In cucumber, improved chilling tolerance, probably through preventing the chill−induced activation of NADPH oxidase and a lower H_2_O_2_ accumulation, was observed [33]. Thus, the decreased ROS accumulation in kiwifruit was probably due to the inhibition of the ROS generation pathway.

Chilling stress induces the production of large amounts of ROS in plants, and the oxidative stress from excessive accumulation of ROS could lead to lipid peroxidation and damage to membrane lipid and protein [6]. The antioxidant system of plants (including antioxidant enzymes such as SOD, CAT, APX, and GR) contributed to removing excess ROS and protecting plants from oxidative damage [7]. It has been proved that increasing the activity of CAT, APX, SOD, and GR activity could effectively control chilling injury and mitigate oxidative damage in horticulture crops. Luo et al. [34] reported that H_2_S treatment reduced the accumulation of ROS by activating SOD, CAT, APX, and GR. According to the present study, PSKα could induce and increase the activities of SOD, CAT, APX, and GR in kiwifruit (Figure 4), thereby inhibiting the accumulation of ROS and maintaining the redox balance of the fruit [30]. As was previously reported, activated SOD, CAT, and APX help to scavenge ROS and therefore improve cold tolerance in many crops, including mandarin fruit [35], cucumber [36], tomato [37], bananas [38], etc. Taken together, these findings supported the hypothesis that PSKα alleviated chilling injury in kiwifruit, probably by inhibiting ROS burst at low temperatures and activating antioxidant systems.

Chilling stress in plants triggers the production of early signal molecules, such as Ca^2+^, NO, ROS, etc. These signal molecules interact with downstream signals, which modulate plant responses to environmental stress [3]. Ca^2+^ plays a crucial role in transmembrane transport and transmission of information, while CaM may act as a receptor or a messenger in the process [13]. Meanwhile, it has become very clear that PSKα is important in plant signaling processes. In recent studies on storage and preservation of horticultural products, it has been found that exogenous calcium could enhance the chilling resistance of fruits and vegetables such as papaya fruit [39], mango [40], bananas [41], and loquat fruit [42] in the process of low-temperature storage; maintain the structure of the cell membranes; prevent the peroxidation of membrane lipids; and maintain the quality of the fruit. PSKα has been confirmed to induce the accumulation of endogenous PSKα and Ca^2+^, which may delay senescence of strawberry fruits during cold storage [24]. Generally, Ca^2+^−ATPase is involved in maintaining intracellular Ca^2+^ homeostasis and Ca^2+^ signaling in plants [42]. Changes in Ca^2+^−ATPase activity are usually influenced by biotic and abiotic stresses [42]. In banana fruit, higher activity of Ca^2+^−ATPase contributed to maintain the energy charge, thus slowing down chilling injury during low-temperature storage [43].

In broccoli florets, PSKα was found to activate Ca^2+^−ATPase, which might potentially ensure sufficient intracellular availability of ATP, thus retarding senescence of broccoli florets during low temperature [26]. The present study indicated that PSKα accelerated CaM accumulation, increased activity of Ca^2+^−ATPase, might lead to an early increase in intracellular Ca^2+^ level, and improved chilling stress resistance in kiwifruit (Figure 5A,B). The mechanism might be due to the activating of the calcium channel by PSKα, a stimulated PSKα−mediated Ca^2+^ signal transduction (CaM, Ca^2+^−ATPase) (Figure 5). In particular, Ca^2+^ may be involved in upstream pathways induced by PSKα. In addition, research in green peppers demonstrated that Ca^2+^ improved chilling resistance by regulating the activity of protective enzymes and membrane lipid composition [44]. Similar results also have appeared in the peach [45], jujube fruit [46], and apricot [47]. Coincidentally, Hou et al. [48] reported that Ca^2+^/CaM contribute to enhancing chilling tolerance in loquat fruit. These findings support our speculations.

MAPK was a common protein kinase of eukaryotes. The MAPK cascade pathway (MAPKKK−MAPKK−MAPK) transmitted signals through sequential phosphorylation and is involved in growth, development, and signal transduction under various biotic and abiotic stresses in plants [19]. Our results suggested that MAPK was activated by PSKα and continued to increase during low-temperature storage. Meanwhile, the expression level of MAPK correspondingly increased, which might be attributed to a cascade reaction. Teige et al. [49] reported that the MEKK1-MKK2-MPK4/MPK6 cascade was involved in the positive regulation of cold response and tolerance, and overexpression of MKK2 led to an increased expression of CBF genes [49]. During low−temperature conditions in *Arabidopsis*, MPK3, MPK4, and MPK6 are rapidly activated. The *mpk3* and *mpk6* mutants exhibit increased expression of CBF genes and improved chilling tolerance. Conversely, constitutive activation of the MKK4/5−MPK3/6 cascade results in reduced expression of CBF genes and increased sensitivity to chilling, suggesting a negative regulation of cold responses by this cascade [20]. Moreover, calcium/calmodulin-regulated receptor kinases, CRLK1 and CRLK2, and MAPKKK have been shown to play an important role in the negative regulation in activating the chilling response of MPK3/6. This suggested that the MAPK cascade and the Ca^2+^ signaling pathway may jointly play an important role in the regulation of cold stress in plants [20].

The above results indicated that PSKα treatment alleviated the chilling injury of kiwifruit and maintained kiwifruit quality by regulating ROS metabolism and activating Ca^2+^ and MAPK signaling pathways. On the other hand, we cautiously postulated that CaM, Ca^2+^−ATPase, and MAPK might be related to ROS generation, and these signaling molecules are participating in chilling responses and play an important role in mediating chilling resistance induced by PSKα during low−temperature storage. In addition, whether the Ca^2+^ signal is involved in salicylic acid (SA), methyl jasmonate (MeJA) or (ABA)-related signaling pathways needs further study.

## 5. Conclusions

Overall, the current study intimated that exogenous PSKα can mitigate the progress of chilling injury in kiwifruit. Membrane completeness was maintained by decreasing the activity of NOX and inhibiting the accumulation of ROS. In addition, PSKα induced the activities and gene expression levels of SOD, CAT, APX, and GR, thereby maintaining redox balance in kiwifruit and providing antioxidant protection against cell damage. Furthermore, the increased levels of CaM and MAPK suggested that Ca^2+^ and MAPK signaling pathways were activated, which might also have contributed to enhancing chilling resistance in kiwifruit. However, given that chilling resistance may be influenced by multiple pathways, further studies are needed to investigate other potential factors and mechanisms.

## Figures and Tables

**Figure 1 foods-12-04196-f001:**
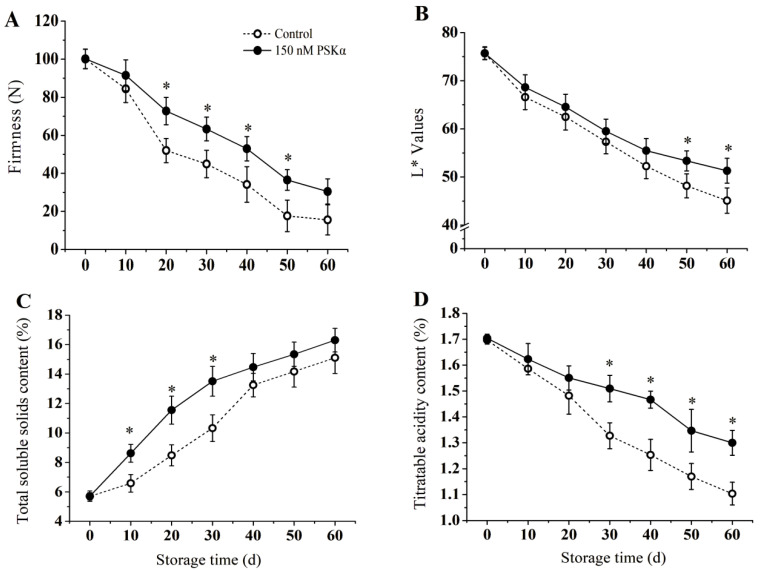
Effect of exogenous PSKα treatment on firmness (**A**), L* values (**B**), TSS content (**C**), and TA content (**D**) of kiwifruit during storage at 0 ± 0.5 °C. Values are presented as means ± SD (*n* = 3). Asterisks (*) indicate significant difference in the figure (*p* <0.05).

**Figure 2 foods-12-04196-f002:**
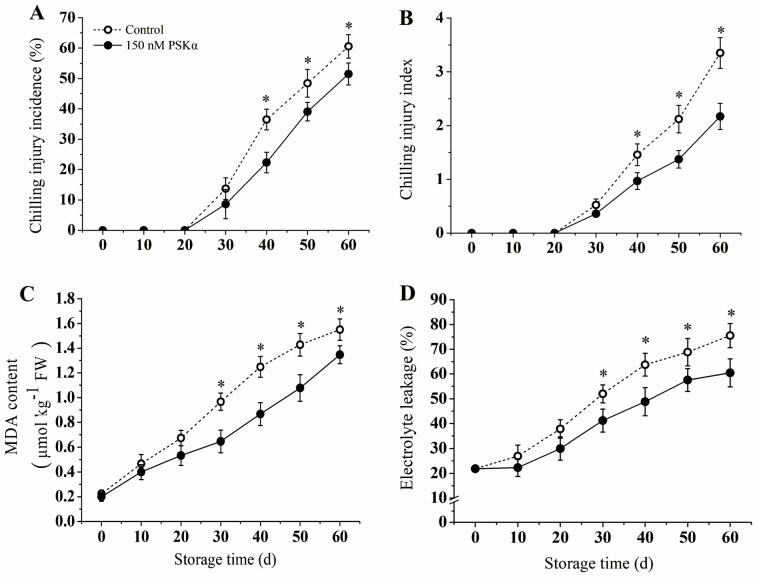
Effect of exogenous PSKα treatment on chilling injury incidence (**A**), chilling injury index (**B**), MDA content (**C**), and electrolyte leakage (**D**) of kiwifruit during storage at 0 ± 0.5 °C. Values are presented as means ± SD (*n* = 3). Asterisks (*) indicate significant difference in the figure (*p* < 0.05).

**Figure 3 foods-12-04196-f003:**
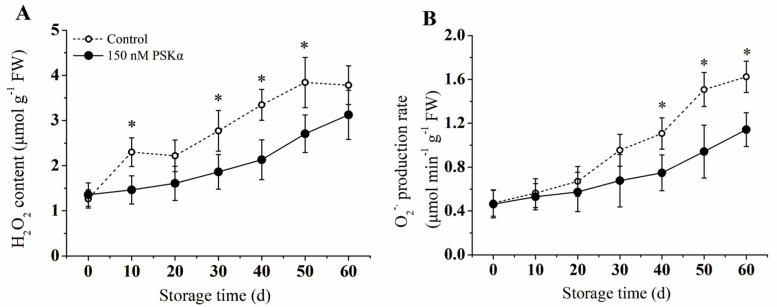
Effect of exogenous PSKα treatment on H_2_O_2_ content (**A**), and generation rate of O_2_^−^ (**B**) of kiwifruit during storage at 0 ± 0.5 °C. Values are presented as means ± SD (*n* = 3). Asterisks (*) indicate significant difference in the figure (*p* < 0.05).

**Figure 4 foods-12-04196-f004:**
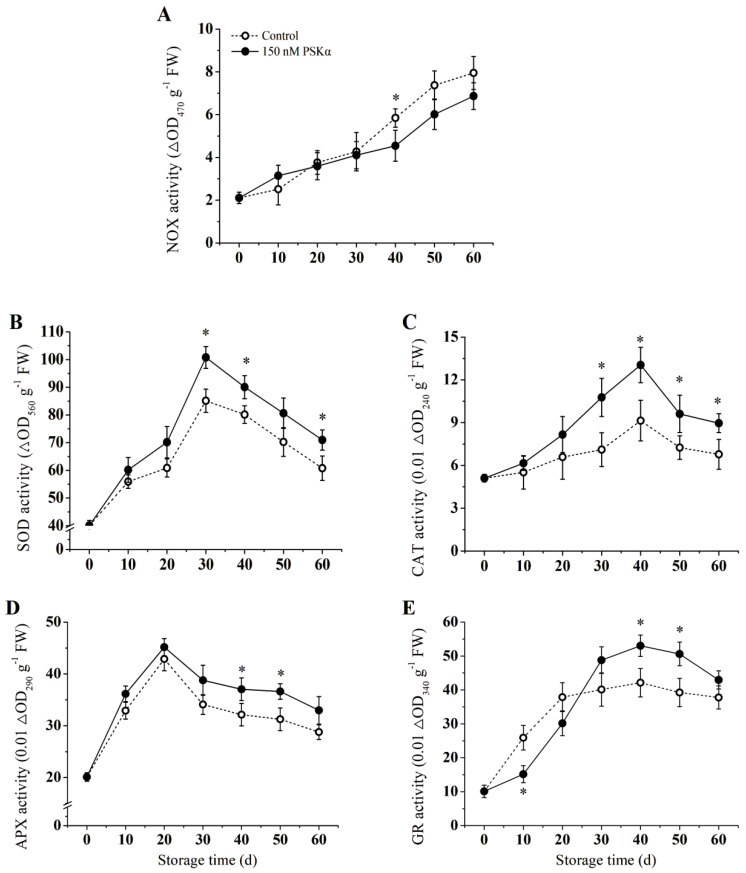
Effect of exogenous PSKα treatment on NOX (**A**), SOD (**B**), CAT (**C**), APX (**D**), and GR (**E**) of kiwifruit during storage at 0 ± 0.5 °C. Values are presented as means ± SD (*n* = 3). Asterisks (*) indicate significant difference in the figure (*p* <0.05).

**Figure 5 foods-12-04196-f005:**
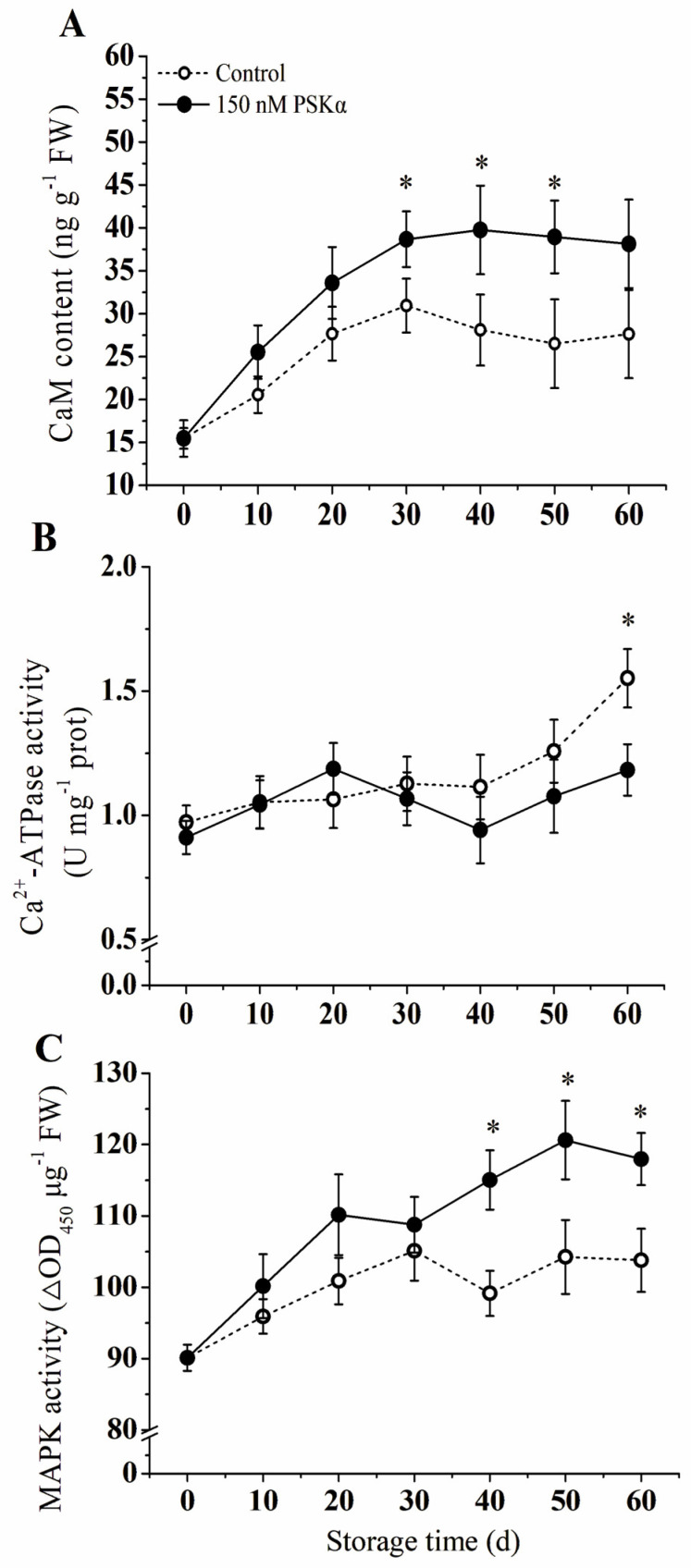
Effect of exogenous PSKα treatment on activities of CaM content (**A**), Ca^2+^−ATPase activity (**B**), and MAPK activity (**C**) of kiwifruit during storage at 0 ± 0.5 °C. Values are presented as means ± SD (*n* = 3). Asterisks (*) indicate significant difference in the figure (*p* < 0.05).

## Data Availability

The data used to support the findings of this study can be made available by the corresponding author upon request.

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
