# Peer review of "Exogenous Phytosulfokine α (PSKα) Alleviates Chilling Injury of Kiwifruit by Regulating Ca^2+^ and Protein Kinase-Mediated Reactive Oxygen Species Metabolism"

_foods, 2023, doi:10.3390/foods12234196_

Round 1
Reviewer 1 Report
Comments and Suggestions for Authors
Line 16: L-1 correct superscript
from start of abstract add two to three of research background review line,
Line 23, subscript " O2- and H2O2"
add key findings end abstarct
remove all small typos related sab script and superscript related, so many typos in whole draft
Line 52; GABA, use full name at first use
Line 60; correct this " O2"
Line 104. botanical name should be italic
Line 155; superscript
Line 166. superscript
add more recent information in discussion section
revision conclusion and remove repeated lines
overall study interesting
Comments on the Quality of English Language
fine
Author Response
Reviewer #1:
- Comment: Line 16: L-1 correct superscript
Response: Thanks to Reviewer for comment. According to reviewer’s suggestion, the unit “-1” was written in superscript in Line 17. Additionally, we checked the whole manuscript and revise the similar kinds of errors. The changes in the text were in red.
- Comment: from start of abstract add two to three of research background review line.
Response: Thanks to Reviewer for comment. The sentence “Kiwifruit fruit stored at low temperatures are susceptible to chilling injury, leading to rapid softening and therefore affecting storage and marketing.” was added in the beggining of abstract section in Line 16-17. The changes in the text were in red.
- Comment: Line 23, subscript " O2- and H2O2"
Response: Thanks to Reviewer for comment. According to reviewer’s suggestion, the words “O2-” and “H2O2” were revised in “O2-” and “H2O2” in Line 25, respectively. The changes in the text were in red. Furthermore , we checked the whole manuscript and revise the typographical errors.
- Comment: add key findings end abstarct
Response: Thanks to Reviewer for comment. According to reviewer’s suggestion, the abstract section was revised as “Kiwifruit fruit stored at low temperatures are susceptible to chilling injury, leading to rapid softening and therefore affecting storage and marketing. The effect of 150 nM mL-1 of exogenous phytosulfokine α (PSKα) on reactive oxygen species (ROS) metabolism, Ca2+ signaling, and signal-transducing MAPK in kiwifruit, stored at 0 °C for 60 days, was investigated. The results demonstrated that PSKα treatment effectively alleviated chilling injury in kiwifruit, with a 15 % reduction in damage compared to the control on day 60. In addition, PSKα enhanced the activities and gene expression levels of superoxide dismutase (SOD), catalase (CAT), ascorbate peroxidase (APX), glutathione reductase (GR), Ca2+-ATPase, and mitogen-activated protein kinase (MAPK). In contrast, the activities and gene expression levels of NADPH oxidase (NOX) were inhibited, leading to a lower accumulation of O2- and H2O2, which were 47.2% and 42.2% lower than those of in the control at the end of storage, respectively. Furthemore, PSKα treatment enhanced the calmodulin (CaM) content of kiwifruit, which was 1.41 times that of the control on day 50. These results indicate that PSKα can mitigate chilling injury and softening of kiwifruit by inhibiting the accumulation of ROS, increasing antioxidant capacity by inducing antioxidant enzymes, activating Ca2+ signaling, and responding to MAPK protein kinase. The present results provide evidence that exogenous PSKα may be taken for a hopeful treatment in alleviating chilling injury and maintaining the quality of kiwifruit.” in Line 16-32. The changes in the text were in red.
- Comment: remove all small typos related subscript and superscript related, so many typos in whole draft
Response: Thanks to Reviewer for comment. We’re sorry for our negligence of not checking writing errors in subscript and superscript. According to reviewer’ s suggestion, we checked the whole manuscript and revise all small typos related subscript and superscript related. The changes in the text were in red.
- Comment: Line 52; GABA, use full name at first use
Response: Thanks to Reviewer for comment. According to reviewer’s suggestion, the full name of GABA was applied. The word “GABA” was revised as “gamma-aminobutyric acid (GABA)” in Line 55. The changes in the text were in red.
- Comment: Line 60; correct this " O2"
Response: Thanks to Reviewer for comment. According to reviewer’s suggestion, the word “O2” was revised as “O2” in Line 64. The changes in the text were in red.
- Comment: Line 104. botanical name should be italic
Response: Thanks to Reviewer for comment. According to reviewer’s suggestion,
the words “Actinidia delicious” were revised as “Actinidia delicious” in Line 109. Furthermore, we checked the whole manuscript and revise the similar kinds of errors. The changes in the text were in red.
- Comment: Line 155; superscript
Response: Thanks to Reviewer for comment. The paragraph “MDA content was determined according to the method of Wang et al. [2]. Fruit pulp tissue (1.0 g) was homogenized in 5 mL of TCA (100 g L-1) solution. The homogenate was centrifuged at 12,000 g for 10 min. The supernatant (2 mL) was collected and mixed with 2 mL of thiobarbituric acid (TBA) (6.7 g L-1), and incubated at 100 °C for 20 min, then cooled rapidly and centrifuged at 12,000 g for 10 min. Absorbance was recorded at 532, 450, and 600 nm. The MDA content was calculated as follows: The absorbance (532 nm) was measured spectrophotometrically by taking 400 µL of the mixture, and the value for non-specific absorption at 600 nm was subtracted. MDA content (µmol kg-1 FW) = [6.45 × (OD532 - OD600) - 0.56 × OD450 ] × Vt × Vr / Vs × m. Where Vt represents total volume of the extraction solution; Vr represents total volume of the reaction mixture solution; Vs represents volume of the extraction solution contained in the reaction mixture solution. m represents mass of the sample based on the fresh weight (FW).” was revised as “According to method of Wang et al. [2], the MDA content was detected. A 2 g of fruit tissue was mixed with 5 mL trichloroacetic acid (TCA) solution (100 g L-1), homogenised, and centrifuged at 10,000 × g for 15 min. The 3 mL of supernatant was collected and mixed with 7.2 g L-1 of TCA solution, the mixture was sealed with plastic wrap and boiled for 20 min. The absorbance was measured at 600nm, 532 nm and 450 nm after cooling. ” Line 157-162. The changes in the text were in red.
- Comment: Line 166. superscript
Response: Thanks to Reviewer for comment. According to reviewer’s suggestion, the unite was written in superscript. The word “ cm2” was revised as “cm2” in Line 164. The changes in the text were in red.
- Comment: add more recent information in discussion section
Response: Thanks to Reviewer for comment. According to reviewer’s suggestion, more recent information about calcium and chilling injury was supplemented in discussion section in Line 379-384. The changes in the text were in red.
- Comment: revision conclusion and remove repeated lines
Response: Thanks to Reviewer for comment. The conclusion section was improved to better summarize the present study. The paragraph “In conclusion, the present suggested that exogenous PSKα treatment can alleviate the development of chilling injury in kiwifruit. The membrane completeness was maintained by decreasing the activity of NOX and inhibiting the accumulation of ROS. In addition, PSKα induced the activities and gene expression levels of SOD, CAT, APX, and GR, maintained redox balance in kiwifruit, which is antioxidant protection against cell damage. Furthermore, the increased CaM and MAPK suggested that Ca2+ and MAPK signaling pathways were activated, which might also be contributed to enhancing chilling resistance in kiwifruit. However, due to the chilling resistance may be under the influence of multiple pathways, further study should be performed to explore other potential factors and mechanisms.” was revised as “Overall, the current study intimated that exogenous PSKα can mitigating the progress of chilling injury in kiwifruit. The membrane completeness was maintained by decreasing the activity of NOX and inhibiting the accumulation of ROS. In addition, PSKα induced the activities and gene expression levels of SOD, CAT, APX, and GR, thereby maintaining redox balance in kiwifruit and providing antioxidant protection against cell damage. Furthermore, the increased levels of CaM and MAPK suggested that Ca2+ and MAPK signaling pathways were activated, which might also have contributed to enhancing chilling resistance in kiwifruit. However, given that chilling resistance may be influenced by multiple pathways, further studies are needed to investigate other potential factors and mechanisms.” in conclusion section in Line 432-441. The changes in the text were in red.
In addition, We have tried our best to polish the language in the revised manuscript. We made some changes in the manuscript. These changes will not influence the content and framework of the paper. And here we did not list the changes but marked in red in revised paper. We appreciate for Editors’ warm work earnestly, and hope that the correction will meet with approval.
Once again, thank you very much for your comments and suggestions.
Reviewer 2 Report
Comments and Suggestions for Authors
· The manuscript has 36 % similarity index indicating high amounts of plagiarised content. The authors are seriously instructed to reduce it as it would create copy right issues for the authors as well as MDPI publishers in future.
· Lane 38-41: Sentence needs rephrasing
· Lane 41: Practical treatment ? sentence appears incomplete pl. check
· Lane 42: numbers of researchers
· Lane 51: Could alleviate chilling injury
· Lane 56: In response to reducing oxidative stress (correct sentence)
· Lane 62-64: sentence is not clear and needs revision
· Lane 79: Dalbergia odorifera needs to be italicised
· Lane 90: Arabidopsis needs to be italicised
· Lane 101: in alleviating chilling injury of kiwi fruit (correct sentence)
· Lane 104: scientific name needs to be italicised
· Lane 105: delete one ‘was’
· Lane 114: Is it 15 or 150 ? Kindly check and correct
· Lane 130-133: For indicating browning, the L-values were considered. Instead if the authors have recorded L, a and b values, they can calculate the browning index using the conversion formula as given in reference: Browning Index (BI) = [100 (x - 0.31)] / 0.17, where x = (a* + 1.75L*) / (5.645L* + a* – 0.3012b* ), according to Ruangchakpet and Sajjaanantakul (2007).
· Lane 134: TA is Titratable acidity
· Lane 166: 1 cm2 (2 in superscript)
· Lane 165: Authors have used the terms relative conductivity in the description while heading has electrolyte leakage which would create confusion for the readers. Hence, it is recommended to use similar terminology at all places.
· Lane 227-230: The description is not matching with the graphical data. Pl. check.
· Lane 262: which were was (word replacement)
· Lane 275-276: The description is not matching with the graphical data. Pl. check.
· Lane 298: Sentence needs revision for proper meaning
· Lane 301-303: 40th day and 60th d………. Please maintain uniformity in depicting days
· Lane 313: during storage
· Lane 316: Sentence needs revision for proper meaning
· Lane 322: check PKSɑ…. It should be PSKɑ
· Lane 329-331: Sentence needs revision for proper meaning
· Lane 336: delete ‘was’
· Lane 337: delete ‘the higher the’
· Lane 339: delete the additional ‘Full stop’
· Lane 343: Sentence needs revision for proper meaning
· Lane 349-350: Sentence needs revision for proper meaning
· Lane 351: replace was with were
· Lane 371: Sentence needs revision for proper meaning
· Lane 372: transmission of information
· Lane 374-376: Sentence needs revision for proper meaning
· Lane 380: low temperature storage.
· Lane 402: Arabidopsis needs to be italicised
· Lane 420: the present study
· Lane 426: also have be contributed
· Lane 427: Sentence needs revision for proper meaning

Needs substrantial improvement. Authors may take the help of professional language expert.
Author Response
List of Responses
Dear Editor and Reviewers,
Thank you for your letter and for the reviewers’ comments concerning our manuscript entitled “Exogenous phytosulfokine α (PSKα) alleviates chilling injury of kiwifruit by regulating Ca2+ and protein-kinase-mediated reactive oxygen species metabolism”. (Manuscript Number: foods-2681000). Your comments are valuable and very helpful for revising and improving our paper, as well as the important guiding significance to our researches. We have studied comments carefully and have made correction which we hope meet with approval. The changes in the manuscript were in red. The main corrections in the paper and the responds to the reviewer’s comments are as flowing:
Reviewer #2:
- Comment: The manuscript has 36 % similarity index indicating high amounts of plagiarised content. The authors are seriously instructed to reduce it as it would create copy right issues for the authors as well as MDPI publishers in future.
Response: Thanks to Reviewer for comment. According to the repetition detection report, the high similarity contents have been revised to reduce repetition rate of the manuscript. The changes are as follows:
(1) The paragraphs “MDA content was determined according to the method of Wang et al. [2]. Fruit pulp tissue (1.0 g) was homogenized in 5 mL of TCA (100 g L-1) solution. The homogenate was centrifuged at 12,000 × g for 10 min. The supernatant (2 mL) was collected and mixed with 2 mL of thiobarbituric acid (TBA) (6.7 g L-1), and incubated at 100 °C for 20 min, then rapidly cooled and centrifuged at 12,000 × g for 10 min. The absorbance was recorded at 532, 450, and 600 nm. The MDA content was calculated as follows:
The absorbance (532 nm) was measured spectrophotometrically by taking 400 µL of the mixture, and the value for non-specific absorption at 600 nm was subtracted.
MDA content (μmol kg-1 FW) = [6.45 × (OD532 - OD600) - 0.56 × OD450 ] × Vt × Vr / Vs × m
Where Vt represents the total volume of the extraction solution; Vr represents total volume of the reaction mixture solution; Vs represents volume of the extraction solution contained in the reaction mixture solution. m represents mass of the sample based on the fresh weight (FW).” was revised as “According to method of Wang et al. [2], the MDA content was detected. A 2 g of fruit tissue was mixed with 5 mL trichloroacetic acid (TCA) solution (100 g L-1), homogenised, and centrifuged at 10,000 × g for 15 min. The 3 mL of supernatant was collected and mixed with 7.2 g/L of TCA solution, the mixture was sealed with plastic wrap and boiled for 20 min. The absorbance was measured at 600nm, 532 nm and 450 nm after cooling. ” in materials and methods section in Line 157-162.
(2) The paragraph “In addition, MAPKKK and two calcium / calmodulin-regulated receptor-like kinases, CRLK1 and CRLK2, negatively regulate cold response activation of MPK3/6.” was revised as “Besides, calcium / calmodulin-regulated receptor kinases, CRLK1 and CRLK2, and MAPKKK have been shown to play an important role in the negative regulation in activating chilling response of MPK3/6.” in discussion section in Line 414-419.
(3) The sentence “In conclusion, the present suggested that exogenous PSKα treatment can alleviate the development of chilling injury in kiwifruit.” was revised as “Overall, the current study intimated that exogenous PSKα can mitigating the progress of chilling injury in kiwifruit. ” in conclusion section in Line 434-435. The changes in the text were in red.
- Comment: Line 38-41: Sentence needs rephrasing
Response: Thanks to Reviewer for comment. The sentence “However, low temperature storage can induce chilling injury (CI) of kiwifruit, which leads to the kiwifruit would be vulnerable to rotting and deterioration when removed from the low temperature [1], and seriously reduced the economic value of kiwifruit.” was revised as “While low temperature storage can effectively inhibit kiwifruit softening and prolong its life, it can also induce chilling injury (CI). Chilling injury can make the fruit vulnerable to rotting and deterioration [1], and significantly reduced the economic value of kiwifruit.” in Line 40-43 for better understanding. The changes in the text were in red.
- Comment: Line 41: Practical treatment ? sentence appears incomplete pl. check
Response: Thanks to Reviewer for comment. The word “practical” was revised as “practical method for”in Line 44. The changes in the text were in red.
- Comment: Line 42: numbers of researchers
Response: Thanks to Reviewer for comment. The phrase “numbers of researchers” were revised as “numerous researchers ” in Line 45 to better express our intention. The changes in the text were in red.
- Comment: Line 51: Could alleviate chilling injury
Response: Thanks to Reviewer for comment. According to reviewer’s suggestion,
The word “alleviating” was revised as “alleviate” in Line 54. The changes in the text were in red.
- Comment: Line 56: In response to reducing oxidative stress (correct sentence)
Response: Thanks to Reviewer for comment. According to reviewer’s suggestion, The phrase “in responding” was revised as “in response” in Line 59. The changes in the text were in red.
- Comment: Line 62-64: sentence is not clear and needs revision
Response: Thanks to Reviewer for comment. The sentence “The ROS-scavening system to prevent chilling stress from developing has been found in many horticulture crops, such as bell pepper [7], okra pod [8], mango fruit [9], and blackberry ” was revised as “The ROS-scavening system have been proven their positive effect to chilling injury in many horticulture crops, such as bell pepper [7], okra pod [8], mango fruit [9], and blackberry [10].” in Line 65-67 for better understanding. The changes in the text were in red.
- Comment: Line 79: Dalbergia odorifera needs to be italicised
Response: Thanks to Reviewer for comment. The word “Dalbergia odorifera” was revised as “Dalbergia odorifera” in Line 83. Furthermore, the whole manuscript was checked and the similar errors were revised. The changes in the text were in red.
- Comment: Line 90: Arabidopsis needs to be italicised
Response: Thanks to Reviewer for comment. The word “Arabidopsis” was revised as “Arabidopsis” in Line 93. The changes in the text were in red.
- Comment: Line 101: in alleviating chilling injury of kiwi fruit (correct sentence)
Response: Thanks to Reviewer for comment. According to reviewer’s suggestion, The phrase “in chilling injury” was revised as “in alleviating chilling injury” in Line 106. The changes in the text were in red.
- Comment: Line 104: scientific name needs to be italicised
Response: Thanks to Reviewer for comment. The word “Actinidia delicious” was revised as “Actinidia delicious” in Line 109. Furthermore, the whole manuscript was checked and the similar errors were revised. The changes in the text were in red.
- Comment: Line 105: delete one ‘was’
Response: Thanks to Reviewer for comment. According to reviewer’s suggestion, the repetitive word “was” was deleted in Line 110. The changes in the text were in red.
- Comment: Line 114: Is it 15 or 150 ? Kindly check and correct
Response: Thanks to Reviewer for comment. To the PSKα-treated and the control groups, the total number of kiwifruit was 300 at each observing time. There were 150 fruits in every group, with three replicates of 50 fruits per group.
- Comment: Line 130-133: For indicating browning, the L-values were considered. Instead if the authors have recorded L, a and b values, they can calculate the browning index using the conversion formula as given in reference: Browning Index (BI) = [100 (x - 0.31)] / 0.17, where x = (a* + 1.75L*) / (5.645L* + a* – 0.3012b* ), according to Ruangchakpet and Sajjaanantakul (2007).
Response: Thanks to Reviewer for comment. According to reviewer’s suggestion, the paragraph “The color change of fruit was measured using a colorimeter (Konica Minolta, CR-400, Tokyo, Japan). Use a whiteboard, correct the colorimeter before testing. The color of the endocarp was measured, and the L* value was recorded, which represents the brightness of the flesh.” was revised as “For indicating browning, the L-values were considered. Instead if the authors have recorded L, a and b values, they can calculate the browning index using the conversion formula as given in reference: Browning Index (BI) = [100 (x - 0.31)] / 0.17, where x = (a* + 1.75L*) / (5.645L* + a* – 0.3012b* ), according to Ruangchakpet and Sajjaanantakul (2007).”. In addition, the relevant reference was supplemented in the References section. The changes in the text were in red.
- Comment: Line 134: TA is Titratable acidity
Response: Thanks to Reviewer for comment. The words “electrolyte leakage” were revised as “Titratable acidity” in Line 139. The changes in the text were in red.
- Comment: Line 166: 1 cm2 (2 in superscript)
Response: Thanks to Reviewer for comment. According to reviewer’s suggestion, the unite was written in superscript. The word “ cm2” was revised as “cm2” in Line 164. The changes in the text were in red.
- Comment: Line 165: Authors have used the terms relative conductivity in the description while heading has electrolyte leakage which would create confusion for the readers. Hence, it is recommended to use similar terminology at all places.
Response: Thanks to Reviewer for comment. According to reviewer’s suggestion, we checked and revised the whole manuscript. The unified expression “electrolyte leakage” was used in the text. The paragraph “The relative conductivity was determined according to the method of Wang et al. [2]. The slices with a diameter (1 cm2) cut from the equatorial region was soaked in deionized water at 25 °C for 30 min. The initial conductivity L0 was determined by the conductivity meter, and the conductivity L1 was determined after boiling for 30 min and cooling to room temperature. The total conductivity is expressed as a ratio of L0 to L1.” was revised as “The electrolyte leakage was determined according to the method of Wang et al. [2]. The slices with a diameter (1 cm2) were cut from the equatorial region and soaked in deionized water at 25 °C for 30 min. The initial electrolyte leakage (L0) was determined using the conductivity meter, and the electrolyte leakage (L1) was determined after boiling for 30 min and cooling to room temperature. The total electrolyte leakage is expressed as a ratio of L0 to L1.” in Line 163-168. The changes in the text were in red.
- Comment: Line 227-230: The description is not matching with the graphical data. Pl. check.
Response: Thanks to Reviewer for comment. The paragraph “TSS content of the control group was significantly higher than that of the PSKα-treated group in the earlier storage period (p < 0.05). On the 30th day, the TSS content of the PSKα-treated group increased by 30.8 % compared with that of the control group (Fig. 1C).” was revised as “In the earlier storage period, the TSS content of the PSKα-treated group was significantly higher than that of the control group (p < 0.05). On the 30th day, the TSS content of the PSKα-treated group increased by 30.8 % compared with that of the control group (Fig. 1C). ” in Line 227-230 to make the description match the graphical data. The changes in the text were in red.
- Comment: Line 262: which were was (word replacement)
Response: Thanks to Reviewer for comment. The word “were” was revised as “was” in Line 263. The changes in the text were in red.
- Comment: Line 275-276: The description is not matching with the graphical data. Pl. check.
Response: Thanks to Reviewer for comment. The paragraph “Compared to PSKα-treated group, NOX activities increased in the control group (Fig. 4A). The NOX activities peaked at 2 h in other treated groups except the control group and then declined and kept a stable active level (Fig. 4A).” was revised as “Compared to PSKα-treated group, NOX activities increased in the control group (Fig. 4A). Fig. 4A also showed there was no significant difference in the change of NOX activity between the control and PSKα-treated groups at the earlier storage. However, after this period, NOX activity continued to increase in the control group. By the 40th day, the NOX activity of the PSKα-treated group was significantly lower than that of the control group.” in Line 274-279 to match the paragraph. The changes in the text were in red.
- Comment: Line 298: Sentence needs revision for proper meaning
Response: Thanks to Reviewer for comment. The paragraph “The treatments of PSKα induced CaM accumulation of kiwifruit during the whole storage, and CaM content reached peak on 40 d, which the activity was 1.41 times higher than that in the control group (Fig. 5A).” was revised as “The treatment with PSKα induced a significant accumulation of CaM in kiwifruit during the whole storage period, and CaM content peaked on day 40, exhibiting a 1.41-fold increase compared to the control group (Fig. 5A). ” in Line 297-299 to better describe the results. The changes in the text were in red.
- Comment: Line 301-303: 40th day and 60th d………. Please maintain uniformity in depicting days
Response: Thanks to Reviewer for comment. According to reviewer’s suggestion, we have checked whole manuscript and the word “day” was used in the text instead of “d”. The changes in the text were in red.
- Comment: Line 313: during storage
Response: Thanks to Reviewer for comment. The sentence “Kiwifruit is susceptible to chilling injury storage at low temperatures” was revised as “Kiwifruit is susceptible to chilling injury during low-temperature storage” in Line 314. The changes in the text were in red.
- Comment: Line 316: Sentence needs revision for proper meaning
Response: Thanks to Reviewer for comment. The sentence “The use of PSKα alleviating chilling injury has been used in various horticultural products [3, 4, 6, 18].” was revised as “The PSKα has been used in alleviating chilling injury in various horticultural products [3, 4, 6, 18]” in Line 317-318 for better understanding. The changes in the text were in red.
- Comment: Line 322: check PKSɑ…. It should be PSKɑ
Response: Thanks to Reviewer for comment. The word “PKSα” was revised as “PSKα” in Line 325. The changes in the text were in red.
- Comment: Lane 329-331: Sentence needs revision for proper meaning
Response: Thanks to Reviewer for comment. The sentence “PSKα treatment reduced CI index and CI incidence of kiwifruit exhibited which ultimately maintained membrane integrity in contrast with the control group (Fig 2A, B)..” was revised as “In addition, Figure 2 shows that during storage, the MDA content, electrical leakage, and cell membrane permeability continuously increased. PSKα treatment reduced CI index and CI incidence in kiwifruit, meanwhile, membrane integrity was maintained in treatment group” in Line 330-333 for better understanding. The changes in the text were in red.
- Comment: Lane 336: delete ‘was’
Response: Thanks to Reviewer for comment. According to reviewer’s suggestion, the word “was” was deleted in Line 339. The changes in the text were in red.
- Comment: Lane 337: delete ‘the higher the’
Response: Thanks to Reviewer for comment. The sentence “Generally, the higher the content of H2O2 and O2-., the stronger the CI incidence has been found in sweet pepper, [25] tomato fruit [26], okra [27], cucumber [28], and pokan fruit [29].” was revised as “Generally, high levels of H2O2 and O2-. could lead to the severe CI, which has been found in sweet pepper, [25] tomato fruit [26], okra [27], cucumber [28], and pokan fruit [29]. ” in Line 340-342 for better understanding. The changes in the text were in red.
- Comment: Lane 339: delete the additional ‘Full stop’
Response: Thanks to Reviewer for comment. According to reviewer’s suggestion, the additional “Full stop” was deleted in Line 342. The changes in the text were in red.
- Comment: Lane 343: Sentence needs revision for proper meaning
Response: Thanks to Reviewer for comment. The sentence “Consistent with previously noted that PSKα delayed senescence of broccoli florets by avoiding ROS accumulation.” was revised as “Previous study have shown that PSKα delayed senescence of broccoli florets by avoiding ROS accumulation, which is consist with present results. ” in Line 346-347 for better understanding. The changes in the text were in red.
- Comment: Lane 349-350: Sentence needs revision for proper meaning
Response: Thanks to Reviewer for comment. The sentence “In tomato fruit, responsible for signaling H2O2 accumulation, NOX activities were found to increase during low temperature in the early stage” was revised as “In tomato fruit, NOX activity was found to increase during low temperature in the early stage, which can be responsible for H2O2 accumulation” in Line 352-353 for better understanding. The changes in the text were in red.
- Comment: Lane 351: replace was with were
Response: Thanks to Reviewer for comment. According to reviewer’s suggestion, the word “was” was revised as “were” in Line 354. The changes in the text were in red.
- Comment: Lane 371: Sentence needs revision for proper meaning
Response: Thanks to Reviewer for comment. The sentence “ These signal molecules interact with downstream signals, leading to responses to stress of plants” was revised as “These signal molecules interact with downstream signals, which modulates plant responses to environmental stress.” in Line 376-377 for better understanding. The changes in the text were in red.
- Comment: Lane 372: transmission of information
Response: Thanks to Reviewer for comment. According to reviewer’s suggestion, the phrase “transmission information” was revised as “transmission of information” in Line 377. The changes in the text were in red.
- Comment: Lane 374-376: Sentence needs revision for proper meaning
Response: Thanks to Reviewer for comment. The sentence “ PSKα has been confirmed to trigger endogenous PSKα signaling, leading to an accumulation of Ca2+ and a subsequent delay in senescence of strawberry fruit during cold storage ” was revised as “PSKα has been confirmed to induce the accumulation of endogenous PSKα and Ca2+, which may delay senescence of strawberry fruit during cold storage ” in Line 384-385 for better understanding. The changes in the text were in red.
- Comment: Lane 380: low temperature storage.
Response: Thanks to Reviewer for comment. According to reviewer’s suggestion, the phrase “low temperature” was revised as “low temperature storage” in Line 390. The changes in the text were in red.
- Comment: Lane 402: Arabidopsis needs to be italicised
Response: Thanks to Reviewer for comment. According to reviewer’s suggestion, the word “Arabidopsis” was revised as “Arabidopsis” in Line 413. The changes in the text were in red.
- Comment: Lane 420: the present study
Response: Thanks to Reviewer for comment. The phrase “the present” was revised as “the current study” in Line 432. The changes in the text were in red.
- Comment: Lane 426: also have be contributed
Response: Thanks to Reviewer for comment. The phrase “also be contributed” was revised as “also have contributed” in Line 438. The changes in the text were in red.
- Comment:Lane 427: Sentence needs revision for proper meaning
Response: Thanks to Reviewer for comment. The sentence “However, due to the chilling resistance may be under the influence of multiple pathways, further study should be performed to explore other potential factors and mechanisms.” was revised as “However, given that chilling resistance may be influenced by multiple pathways, further studies are needed to investigate other potential factors and mechanisms. ” in Line 440-442 for better understanding. The changes in the text were in red.
- Comment: Needs substrantial improvement. Authors may take the help of professional language expert.
Response: Thanks to Reviewer for comment. We are very sorry for our poor English writing. The manuscript has been revised and re-polished by a native English speaker. We made some changes in the manuscript. These changes will not influence the content and framework of the paper. And here we did not list the changes but marked in red in revised paper.
In addition, We have tried our best to polish the language in the revised manuscript. We made some changes in the manuscript. These changes will not influence the content and framework of the paper. And here we did not list the changes but marked in red in revised paper. We appreciate for Editors’ warm work earnestly, and hope that the correction will meet with approval.
Once again, thank you very much for your comments and suggestions.
Yours sincerely,
Qingjun Kong
E-mail: wangdi237@163.com
Reviewer 3 Report
Comments and Suggestions for Authors
In the manuscript titled 'Exogenous phytosulfokine α (PSKα) alleviates chilling injury of kiwifruit by regulating Ca2+ and protein-kinase-mediated reactive oxygen species metabolism', the authors provide significant results about the activity of the exogenous PSKα for the treatment in alleviating chilling injury and maintaining the quality of kiwifruit. Only some technical errors (e.g. line 16: -1 should be in superscript, line 17 2+ should be in the superscript, line 19: 15 % should be 15% Line 23: O2 and H2O2 2 should be in subscript, etc.) mean that a minor revision of the whole manuscript is needed. Please check the whole manuscript and revise the typographical errors.
Comments on the Quality of English LanguageThis manuscript is, in my opinion, appropriate for such a reputable journal after minor mentioned revisions.
Author Response
List of Responses
Dear Editor and Reviewers,
Thank you for your letter and for the reviewers’ comments concerning our manuscript entitled “Exogenous phytosulfokine α (PSKα) alleviates chilling injury of kiwifruit by regulating Ca2+ and protein-kinase-mediated reactive oxygen species metabolism”. (Manuscript Number: foods-2681000). Your comments are valuable and very helpful for revising and improving our paper, as well as the important guiding significance to our researches. We have studied comments carefully and have made correction which we hope meet with approval. The changes in the manuscript were in red. The main corrections in the paper and the responds to the reviewer’s comments are as flowing:
Reviewer #3:
- Comment: In the manuscript titled 'Exogenous phytosulfokine α (PSKα) alleviates chilling injury of kiwifruit by regulating Ca2+and protein-kinase-mediated reactive oxygen species metabolism', the authors provide significant results about the activity of the exogenous PSKα for the treatment in alleviating chilling injury and maintaining the quality of kiwifruit. Only some technical errors.g. line 16: -1 should be in superscript
Response: Thanks to Reviewer for comment. According to reviewer’s suggestion, the unit “-1” was written in superscript in Line 17. The changes in the text were in red.
- Comment: line 17 2+ should be in the superscript,
Response: Thanks to Reviewer for comment. According to reviewer’s suggestion, the unit “2+” was written in superscript in Line 18. The changes in the text were in red.
- Comment:line 19: 15 % should be 15%
Response: Thanks to Reviewer for comment. According to reviewer’s suggestion, a space was deleted between “15” and “%” in Line 21. The changes in the text were in red.
- Comment:Line 23: O2 and H2O2 2 should be in subscript, etc.) mean that a minor revision of the whole manuscript is needed. Please check the whole manuscript and revise the typographical errors.
Response: Thanks to Reviewer for comment. According to reviewer’s suggestion, the words “O2” and “H2O2” were revised in “O2” and “H2O2” in Line 25, respectively. The changes in the text were in red. Furthermore , we checked the whole manuscript and revise the typographical errors.
In addition, We have tried our best to polish the language in the revised manuscript. We made some changes in the manuscript. These changes will not influence the content and framework of the paper. And here we did not list the changes but marked in red in revised paper. We appreciate for Editors’ warm work earnestly, and hope that the correction will meet with approval.
Once again, thank you very much for your comments and suggestions.
Yours sincerely,
Qingjun Kong
E-mail: wangdi237@163.com